# Interfaith Chaplaincy as Interpretive Hospitality

**Peter Ward Youngblood**

Department of Cultural and Religious Studies, The Chinese University of Hong Kong, Sha Tin, NT, Hong Kong, China; pwyoungblood@link.cuhk.edu.hk

**Abstract:** Hospital chaplaincy must reconcile competing epistemologies of health and salvation (Christian, clinical, holistic, etc.), but when done in interfaith situations this task becomes more difficult. As current models of spiritual care are insufficient, this paper proposes a paradigm based on Paul Ricoeur's hermeneutics of translation, as adapted for comparative theology by Marianne Moyaert. In particular, it looks at his idea of linguistic hospitality as a way to structure relations, spiritual assessments, and pastoral interventions in interfaith chaplaincy without reducing the unique strangeness of "the Other". Furthermore, a practical, performative (ritual) hospitality can overcome the epistemological and soteriological obstacles that have frustrated systematic theologies of religion.

**Keywords:** chaplaincy; interfaith; comparative theology; hermeneutics; Ricoeur; interreligious dialogue

## 1. Introduction

An interfaith chaplain is a spiritual caregiver who provides emotional and spiritual support to all persons, of any religion. In doing so, she must balance the various and competing cultural and religious norms that define human well-being. Therefore, spiritual care always involves several tensions: First, a chaplain must respect the patient's[1] beliefs while also being authentic to her *own* tradition. Second, she must provide religious services in a secularized profession operating in secularized institutions (hospitals). Finally, she must account for other conflicting social and cultural factors, including family, ethnicity, and community.

Such tensions can be creative and rewarding, but negotiating these conflicting epistemologies is no easy task, and to do so in a culturally and religiously diverse place like East Asia is even more challenging. My research context is Hong Kong, where—as in the U.S.—Christian chaplains are normally expected to provide care to all persons. But not only are there many diverse religions to keep track of, the boundaries between them are often unclear. In such contexts, Western paradigms and practices of interfaith care are not up to the task. Chaplains tend to defer to either pluralist metaphysics or generic language (or both), uncritically avoiding the problem of interreligious conflict. Instead, I believe comparative theology and its moral paradigm of interreligious *hospitality* would be more useful.

In this article, I will develop a Christian spiritual/pastoral[2] care method that uses hospitality as a hermeneutical (the theory and method of interpretation) ethic, drawing primarily from Marianne Moyaert and her use of Paul Ricoeur's (1913–2005) "linguistic hospitality." I will describe the ways hospitality has been recovered as a Christian tradition and applied to interreligious dialogue, and then I will summarize Moyaert's development and use of linguistic hospitality in comparative theology.

---

1    By "patient" I mean not just hospitalized persons, but also anyone else who may need care, i.e., family and staff.
2    "Spiritual" is often considered the more inclusive term, but "pastoral" is still frequently used, even by non-Christian practitioners.

Any discussion of spiritual care and hermeneutics cannot ignore the late pastoral theologian Donald Capps' (1939–2015) research in this area, so I will also draw upon his own approach to Ricoeur. After briefly discussing the complexities of religion and spiritual care in Hong Kong, I will show how interreligious hospitality can be a more suitable theological and ethical paradigm for interfaith work, especially in diverse contexts.

## 2. Developments in Comparative Theology and Studies of Hospitality

Comparative theology is "faith seeking understanding" through dialogue with other religions. This stands in contrast to theologies of religious pluralism or "theologies of religions" (TORs), Christian responses to the epistemological and soteriological challenges of other faiths. Behind such responses are two key questions: (1) how do we judge the truth-claims of other religions and (2) more importantly, do other religions *save*? The classical TOR typology of *exclusivism*, *inclusivism*, and *pluralism*, originally developed by Race (1983), has been heavily debated. "Pluralists" generally accept that other religions can be as true and salvific as Christianity, while "inclusivists" assign a higher status to the latter. Exclusivists generally reject the possibility of truth and salvation outside of Christianity or "the Church" (however that may be defined). Pluralist or more progressive inclusivist theologies are favored by mainstream theologians as these are seemingly more tolerant and open to dialogue. However, as *a priori* positions *all* such theologies are *exclusive* in some way; Even pluralists must logically exclude more conservative exclusivist positions (i.e., those that reject pluralism). While as normative systems such theologies can be intolerant or hostile, they do describe the full spectrum of theological positions that a scholar or practitioner will bring with them into any interreligious relationship (Yong 2008).

Comparative theology avoids the problem of exclusivity because its truth-claims arise *a posteriori* through the dialogical process, not before. The only things that must precede dialogue are considerations of practical method, relationship-building, and—of course—ethics. For comparativist Marianne Moyaert, "comparative theology is moved by an ethical concern to enable a just relation between both *one's own tradition* and the *foreign* one" (Moyaert 2012, p. 1145). Because every theologian will have their own "theology of religion" that will influence their approach, a practical-ethical standard is needed. However, the problem is that the basis for any such ethics would *also* be tradition-mediated. For this reason, ethicist Luke Bretherton is skeptical that there can be an objective form of meta-ethics that resolves interreligious moral disputes.[3] Instead, Christians and other religious groups need their own theologically-grounded practical-ethical paradigms with which to structure their relationships with each other (Bretherton 2006).

This need has led Bretherton, Moyaert, and other Christian scholars to the concept of *hospitality*. Derived from the Latin *hospes* and called *philoxenia* (love of strangers) in the New Testament Greek, hospitality is not only a central Christian tradition but also a basic principle of human interaction. Obviously this kind of hospitality is more than domestic etiquette or the "hospitality industry". Rather, here it refers to a philosophical and ethical concept central to human relationships and a source for a radical Christian ethic. Though I will simply define it as the "the act of receiving a stranger", hospitality is a complex concept, loaded with ambiguity and paradox and containing a broad lexicon of related terms, such as "host", "guest", "stranger", and "gift".

Hospitality as a radical ethic is based heavily on the philosophical work of Emmanuel Levinas (1905–1995) and Jacques Derrida (1930–2004). Levinas focused on the phenomenology of "the Other"—how we encounter other persons, how we are responsible for them, and how this responsibility interrupts and transforms our own self-identity.[4] Influenced by Levinas, Derrida saw hospitality not just as the core principle of his late friend and mentor's work, but the very foundation of ethics itself (Derrida 1999). He further argued that true hospitality is "impossible", as it must be completely

---

[3]    In particular, Bretherton critiques ethicist Alisdair MacIntyre's approach.
[4]    For Levinas' phenomenology of "the Other", see *Totality and Infinity* (Levinas 1991).

unconditional: the guest must be able to come unbidden without invitation or limit and remain a total stranger, i.e., "wholly other", while the host is effectively the guest's hostage. In reality, such hospitality is not possible as conditions and limits must always be set, these often imposed forcefully or violently through rules, payments, and laws and punishment. However, these conditions or limits—what Derrida called the "laws of hospitality"—are the very things that make hospitality possible (Derrida and Dufourmantelle 2000; Wrobleski 2012). For example, I cannot offer my home to someone unless I know something about them (who they are, what they may need, etc.) and rules are required to ensure our mutual safety. These conditions mean that hospitality in the truest sense of the word is not possible. Despite this paradox, Derrida presented unconditional or absolute hospitality—what he called the (capital "L") Law of Hospitality—as a guideline for being as hospitable as possible (Derrida and Dufourmantelle 2000).

## 3. Theologies of Hospitality

According to ethicist Christine Pohl, hospitality was an early Christian tradition that was lost over time to the domestic sphere and specialized institutions like homeless shelters and hospitals (Pohl 2011). But in the past half-century, hospitality has received renewed attention. It was, of course, a crucial theme for Henri Nouwen (1932–1996) and his spiritual and pastoral ethics.[5] More recent comprehensive works include those by Pohl (1999); Bretherton (2006); Russell (2009); and Wrobleski (2012). Many of these scholars developed their own "theologies of hospitality", paying special attention to the concept's philosophical and ethical implications. In *Hospitality as Holiness*, Bretherton envisions hospitality as the theological framework for resolving moral diversity (Bretherton 2006). In *Hospitality and the Other*, Pentecostal theologian Amos Yong draws primarily from Derrida, arguing that the solution to hospitality's "impossibility" is a superabundant divine hospitality that spreads outward through the "many tongues and many practices" of the Pentecost event (Yong 2008). Explorations of hospitality are not restricted to Christianity; A religiously-diverse group of scholars contributed to *Hosting the Stranger: Between Religions* (Kearney and Taylor 2011), in an effort to show that a sacred commitment to the stranger is shared by all major faiths.

Most Christian theologies of hospitality share several common themes: All agree that *absolute* hospitality can be provided only by the triune God who invites all human beings to His heavenly banquet table (e.g., Luke 14). In doing so, He also commands our own hospitable acts in which we are to receive others, including absolute strangers. To host strangers is to also host God, Christ, or angels, figuratively or literally. Abraham's divine encounter at Mamre (Gen 18) inspired later exhortations like Hebrews 13:2 ("entertaining angels without knowing it").[6] The roles of host and guest/stranger are never static, but ambiguous and alternating. Abraham and his descendants first lived as sojourners and refugees but later became rulers and hosts in Palestine. Jesus Christ was simultaneously guest and (divine) host throughout his ministry, being invited into many homes only to reveal God's hospitable presence through his teachings and miracles.

Due to our fallenness or finitude (and in keeping with Derrida's paradox), human beings can never unconditionally accept "the other", but rather our efforts at hospitality turn into attempts at conformity, control, and even hostility. It is our personal, communal, and ethnic need for safety and security that shapes our compassion and not the other way around. We establish hierarchies of guests, opening our doors only to those we know well or who can offer us something in return. Hebrew law contains many demands to accept the stranger (*ger*), reminding the Israelites that they were once "strangers in Egypt" (Ex 22:21; Deu 10:19), but their exilic and diasporic experience led rabbinical scholars to redefine *ger* as a converted Jew, rather than the cultural and religious outsider (Goshen-Gottstein 2014).

---

[5] For a good example of Nouwen's contemplative hospitality, see *Reaching Out: Three Movements of the Spiritual Life* (Nouwen 1986).

[6] All Biblical references are taken from the New Revised Standard Version (NRSV).

Christ reversed contemporary notions of Jewish purity by making hospitality a condition for holiness and not vice-versa (Bretherton 2006), but Christian communities have since turned more xenophobic and exclusivist in many places. Hospitality has consequently been "institutionalized"—divided among secular social and commercial services.

Theologians of hospitality believe overcoming this problem requires an eschatological perspective and a robust pneumatology that distinguishes Christians from non-Christians without excluding them spiritually. God has redeemed creation through Christ, but differences remain because of the "not yet" of Christian eschatology. Before Christ's return, there exists continuity and discontinuity between non-Christians and Christians, the latter living simultaneously between the historical present and the coming Kingdom of God (Bretherton 2006). Christians are privileged, but only in their knowledge of this Truth. They are not morally and spiritually superior to non-Christians, who can still be led to God through the Holy Spirit.

Theologically, this is a form of *inclusivism*, though the emphasis is not on theological truth-claims but rather the hospitality that arises through our responses to Christ and the Holy Spirit. This importance of *praxis* is evident to Yong, who believes the incommensurability of TORs may be overcome by focusing on their practices, not their theological claims. While divided in their cognitive beliefs, pluralists, inclusivists, and exclusivists perform these beliefs in many ways, some of which are similar in their expressions of hospitality (Yong 2008). Forms of evangelism may even be seen as hospitable practices, as long as they are done with humility, respect, and empathy. Hospitality has already been explored as a new paradigm for mission (e.g., Brandner 2013). Through such hospitable practices, the Spirit cuts across all cultures and religions, places and times, bringing humanity and the rest of Creation together. While theological tensions remain, hospitality functions as a transformational ethic that enables care rather than conflict. It is transformational insofar as it challenges the Christian's deeply-held views (TORs) that often encourage hostility.

## 4. Interreligious Hospitality

Now I will concentrate on a "linguistic approach" to interreligious hospitality developed by Marianne Moyaert, who uses the text hermeneutics of Ricoeur. In *Fragile Identities: Towards a Theology of Interreligious Hospitality* (2011), she challenges TORs, particularly a one-sided interpretation of George Lindbeck's (1923–2018) cultural-linguistic model of religion that she calls *particularism*. In the cultural-linguistic model, religions, like languages, are self-contained systems of meaning governed by their own rules (i.e., grammar). A "particularist" believes that these systems are so unique that they are unknowable (untranslatable) by those from other religions, which makes meaningful interreligious dialogue impossible. While she affirms that cultural-linguistic models are useful, Moyaert rejects this narrow view, arguing instead for a form of comparative theology that remains open to the strangeness of other religions. Theologically, Moyaert grounds her argument in a reinterpretation of the Story of Babel (Gen 11), which she views not as humankind's "second fall", but as an act of God affirming cultural and linguistic diversity. Babel was the beginning of linguistic *polysemy*—in which words or phrases have many different meanings (Moyaert 2011).

Linguistic polysemy brings Moyaert to Paul Ricoeur and his hermeneutical theory of translation. Ricoeur's hermeneutics is premised on the idea that we can never truly know the original, authorial meaning of a text[7] and, similarly, there can be no such thing as a perfect translation. Because interpretation and translation are never direct, they instead open up a new world of meaning between the reader and the text, enabling creativity and innovation. Even though something is always lost in translation, something is also gained. However, some fidelity is still expected, especially in the case of

---

7　Ricoeur's interpretation theory cuts across much of his extensive bibliography, but see in particular his *Interpretation Theory: Discourse and the Surplus of Meaning* (Ricoeur 1976). His work on the hermeneutics of translation may be found in *On Translation* (2006). I am not an expert on Ricoeur's thought and so my summary here and in the next section is indebted to Moyaert's (2011) explanation as well as Capps (1984).

translation work, where there is less spatiotemporal distance between the translator and the author of the text. This creates a tension, as the translator must be faithful to both the original language and their own language (Moyaert 2011; Ricoeur 2006).

To overcome this tension, Ricoeur recommended *linguistic hospitality*, "where the pleasure of dwelling in the other's language is balanced by the pleasure of receiving the foreign word at home . . . " (Ricoeur 2006, p. 10). Like Derrida's hospitality, this hospitality accepts the "otherness" of the stranger/strange language, resisting the desire to make it more familiar. Instead the host/translator allows the unfamiliarity of the strange text to challenge them. However, *both* languages will inevitably be changed in some fundamental way. The translator will try as hard as possible to keep the uniqueness of the original tongue, but she must still conform it to another language. Likewise, the host language is transformed by the foreign word or meaning, as the latter can never fit comfortably into the former—it will *add* something. Unlike Derrida, Ricoeur believed that the translator's inevitable double failure is a *good* thing, since it allows a new world of meaning to arise and contribute to the host language (Moyaert 2011; Taylor 2011). To give a rough example of a kind of "linguistic hospitality", in my department it is not uncommon to suddenly hear words like "transcendence" or even "hermeneutics" thrown into a Cantonese conversation. Having no satisfactory Chinese equivalents, these and other more common English terms were borrowed. Today they may sound natural to Hong Kongers, but peculiar to native speakers like me.

Linguistic hospitality can be understood as balancing—albeit imperfectly—the tension between familiarity and alterity in translation. Moyaert believes this metaphor is apt for comparative theological work. Both translation and comparative theology take place in a "fragile hermeneutical space—fragile because every interpreter finds herself in a field of tension between faithfulness and betrayal" (Moyaert 2011, p. 236). A comparative theologian must practice their own form of hospitality, or "hermeneutical openness", that exposes their belief system to the strangeness of another's, risking its transformation. Comparative theology is thus a "vulnerable theology": it requires the vulnerability of the one being interpreted, who could be misinterpreted, and the vulnerability of the theologian, whose belief system could be similarly affected (Moyaert 2011, 2012, 2014).

## 5. Hermeneutics and Spiritual Care

But what has comparative theology to do with spiritual care and chaplaincy? Like Ricoeur's hermeneutics, comparative theology normally focuses on religious *texts*, not conversations with persons. Even if we extend the field to include all interreligious dialogue, such work is still normally for academic purposes or wider social issues like religious violence and immigration. Little has been discussed about the intimate, interpersonal level of dialogue where spiritual care takes place. I believe comparative theology and interreligious hospitality have something to add as they (1) provide a practical hermeneutical ethic and (2) take real religious differences seriously.

Capps (1984) recognized the usefulness of hermeneutics for pastoral care, noting that it was already involved in the "action-reflection-action" model of Clinical Pastoral Education (CPE) developed by chaplain Anton Boisen. CPE is the educative basis of chaplaincy and other forms of pastoral/spiritual care, enabling contextual pastoral reflection and formation regarding the relationship of theology with the biological and psychosocial dimensions in particular. Capps sought to develop a more rigorous, structured, and detailed model of pastoral care hermeneutics using Ricoeur's hermeneutical theories as a foundation. While Ricoeur usually wrote about interpreting texts, he also believed there could be an interpretation of "meaningful actions" that (1) have significance or influence beyond the moment of their occurring, (2) have unintended consequences, and (3) are open to interpretation (Capps 1984; Ricoeur 1984). Capps applied Ricoeur's textual model to meaningful pastoral actions (Capps 1984). Such actions could arguably be most any kind of intervention, such as bereavement counseling, family mediation, ritual services, or even casual chaplaincy visits.

Capps believed pastoral actions can be like texts because they have an immediate meaning to the "authors" of the action—caregiver and care-receiver—which is then superseded by a higher

level of meaning that arises between the action and a future interpreter of the action. Unlike textual interpretation, the "authors" are *also* the interpreters of pastoral actions, as they are interpreting something they did in the past. This creates what Ricoeur called *distanciation* between the persons and their actions, which allows reflection and re-reflection on the event in order to reveal new meanings. Hermeneutical reflection always involves an initial guess of the meaning (*pre-understanding*), objective analysis and validation (*explanation*), and a new, reoriented understanding which is then appropriated into the pastor's work (*appropriation*) (Capps 1984). In spiritual care, reflection is primarily the task of the pastoral counselor or chaplain (and possibly their colleagues, supervisors, and other relevant staff), but they *must* involve the patient at some point.

To illustrate this process, I can provide a simple example from my own CPE experience in the US. Many patients would often say "it is what it is" when I mentioned their life-threatening, chronic, or terminal illnesses. At first I always left these conversations frustrated; Such statements by the patient seemed to close-off conversation, as if their serious illnesses were not an important topic or they were in denial. But post-visit reflection with my colleagues and CPE supervisor revealed alternative, broader *understandings* of what could have happened. For instance, the patient could have been perfectly accepting of the situation, requiring no further spiritual support. Alternatively, they could have been sadly resigned or even too scared to talk about it. It was also important to not just focus on that one statement, but the whole "action" (encounter), including my own part in it. The patient may have simply been uncomfortable speaking to *me*—maybe I had not done enough to establish a safe space or present myself as a reliable listener. The chaplain's own humility and vulnerability are critical elements of spiritual care hermeneutics.

The flip-side of this dynamic is that the chaplain's understanding of a pastoral encounter or intervention will have its own "world" of meaning, which could still be valid for the chaplain and others, even if it does not accord with the patient's view. Such *understandings* are, of course, still guesses and need to be defended publically (*explanation*). Furthermore, the critical difference between textual interpretation and pastoral/spiritual care is that the latter has a therapeutic aim—the patient needs support and healing, and so their experience takes priority. The chaplain's subjective understanding and explanation of meaning would need to be validated in collaboration with the patient. Therefore, unlike Ricoeur's text hermeneutics, greater fidelity to the patient's "authorial" meaning is necessary. Upon validation or correction, a chaplain can integrate (*appropriate*) this new understanding into their continued care of that patient (or for other patients in similar situations).

## 6. Hermeneutics and Interfaith Care

### 6.1. Post-Christian and Interfaith Approaches

Per Ricoeur, interpretation is a dynamic, transformational, and cyclical (though not viciously so) process of (pre-) *understanding*, *explanation*, and (revised and appropriated) *understanding* similar to "action-reflection-action." But these processes remain abstract, lacking a clear and concrete theologically-informed ethics. On the one hand, the intersubjective space between chaplain and patient—and the hermeneutical space between them and the "pastoral action"—functions as a neutral zone where particular spiritual beliefs and needs may be expressed and discussed. On the other hand, the hermeneutical process, structure, and space itself will be preemptively shaped according to the embedded theological or spiritual norms of chaplain and patient (whether they realize this or not). This is a bias built into the very questions that chaplains ask as part of spiritual assessments (Balboni 2013).

Anticipating the need for greater objectivity, Capps (1984) proposed three inclusive, correctable explanatory schemata to guide pastoral praxis and self-understanding, based on models already well-developed in pastoral theology: "shepherd", "wise fool", and "wounded healer". Over time more paradigms have been proposed. For instance, Dykstra's (2005) book *Images of Pastoral Care* includes Boisen's "living human document", Capps' updated "wise fool", plus seventeen others. While useful, such paradigms are usually put forth by Christian authors using predominantly Christian metaphors

aimed at *intra*-Christian pastoral care. At the same time, some, such as Bonnie Miller-McLemore's "living human web" (an adaptation of Boisen), more greatly emphasize the importance of the social context (Miller-McLemore 2005). Such approaches are better fits for family- and community-oriented societies like China and Hong Kong.

These revised schema or paradigms reflect the gradual deconstruction of the Protestant Christian-centric approach to spiritual care through postmodern critique (i.e., racial, ethnic, gendered, socioeconomic, etc.). But while this helps identify and overcome bias, such approaches tend to eschew normative truth-claims in order to maintain total inclusivity. As a result, the deeply-held and conflicting theological positions of individuals are glossed-over or ignored. There has been a notable lack of attention to *interfaith* spiritual care and its theological implications (Greider 2012). The paradigms most commonly used (often unconsciously) are humanistic and pluralistic metaphysical frameworks based on the understanding that all human beings are spiritual and express this spirituality in many different ways.[8] Christian interfaith chaplains, worried about being too direct or proselytizing, will often avoid talking about their own religion and instead try to use the spiritual language, symbols, and practices of the patient. Alternatively, they may deploy neutral or experiential language (hope, meaning, transcendence, etc.) that appeals to a universal common ground or "meta-theology" of sorts. Sociologists call these strategies *code-switching* and *neutralization*, respectively (Cadge and Sigalow 2012).

Code-switching and neutralization could be understood as forms of translation, but they are problematic ones. There are few meaningful "generic" spiritual services the Christian chaplain can provide the Muslim patient. While the neutralizing chaplain might think that "everyone is spiritual" this would ignore a Muslim's very specific and concrete religious needs, such as ritual washing and daily prayers (Abu-Ras and Laird 2011). Worse, code-switching cuts very close to cultural or religious appropriation, even if the intent is to help. A Christian cannot, for instance, recite the *Shahada*[9] without the commitment to *become* a Muslim. This is why Moyaert thinks Ricoeur's third step of "appropriation" is better understood as *expropriation*. The former implies "interpretive imperialism", while the latter better expresses a linguistic hospitality where the other religion is kept unfamiliar while the theologian's religious beliefs and practices are "redescribed and transformed" (Moyaert 2017, p. 180).

Another issue is that the "generic" or "secular" concepts and language used as part of neutralization carry embedded biases, usually Western-Christian ones. This includes the contained, independent, and integrated notion of the self that is prevalent in the "patient-centered" therapeutic approaches still popular among chaplains. Such thinking is a problem for the cultural contexts where individualism is often superseded by familial and communal relationships, as in Hong Kong. Furthermore, most secularized or generalized understandings of spirituality maintain a body-spirit dualism that is also incompatible with Chinese culture (Kwan 2018).

In code-switching and neutralization, the epistemological, anthropological, soteriological, and—importantly—ethical norms are established from outside the religious language-systems meaningful to either the patient, chaplain, or both. Ricoeur admitted some betrayal will happen in any translation, but the translator still actively tries to satisfy to both sides (hence the creative tension). In contrast, the above strategies actually represent an *intentional* or *uncritical* betrayal of the patient's and/or chaplain's cultural-linguistic-religious systems. Code-switching risks uncritically adopting the religious language of the patient at the expense of the chaplain's own, and to neutralize is to defer to external, humanistic, or abstracted norms that may have little meaning for either chaplain or patient. This relates to the philosophical tension of particularity versus universality inherent in chaplaincy work.

---

[8]　I do not want to suggest that such understandings are always unhelpful or uncritical, but they often understate the real differences and potential conflicts in interreligious dialogue, especially those taking place between more conservative religious persons.

[9]　"There is no God but God (Allah), and Muhammed is the messenger of God."

*6.2. The Chinese Way of "Doing Religion"*

Finally, modern spiritual caregiving does not give enough attention to the complexity and diversity of religious life in China. An American or European chaplain well-versed in Buddhist, Daoist, or Confucian thought—collectively called the *san jiao* ("three teachings")—would still be surprised by the diverse beliefs and practices encountered in Hong Kong. These include theistic and devotional practices as well as popular practices pejoratively associated with magic or "superstition".

Unlike bounded traditions such as Christianity, Judaism, and Islam, Chinese religions are inclusive or syncretic, highly-localized, and diffused into everyday life. Practicing cross-religiously between Buddhist, Daoist, and Confucian practices is common due to pragmatic and selective participation in a wider spiritual system that predates the *san jiao*. This ancient system includes customs such as divination, *feng shui*, ancestral veneration, and deity worship and is often conflated with Daoism or categorized as "popular religion".

Anthropologist Adam Yuet Chau prefers to think of *modalities* of "doing" Chinese religion that cut across the often-arbitrary institutional and cultic divisions. Among these, the "discursive/scriptural" and "personal-cultivational" modalities relate to the educated, elite expressions that Westerners are more familiar with. These typically focus on the texts and philosophies/theologies of the *san jiao* as well as the moral and physical cultivation practices popular in Western countries (e.g., mindfulness meditation, *tai chi*). Less well-known in the West are the "liturgical", "immediate-practical", and "relational" modalities better representative of lay Chinese practices in Hong Kong. These pertain to the highly pragmatic devotional, ritual, and magical interactions that persons have with the whole cosmos of gods, ghosts, and ancestors (Chau 2011).

## 7. Are There Already Chinese Models of Spiritual Care?

Given this diversity of belief and practice, one question is whether there is already a form of Chinese spiritual care that Christian practitioners may draw upon. However, despite the continued vitality of traditional Chinese religion in Hong Kong, for several reasons pastoral/spiritual care remains a predominantly Christian-based practice. First, Christians have had an outsized influence in Hong Kong society despite their minority status. Christian missionary organizations and communities established and/or funded many of the city's schools, hospitals, and other social services in the former British colony. Today, Christianity is an "elite" religion, practiced by many businesspeople, politicians, and other local leaders.

Second, Chinese religious services are not widely available in hospitals.[10] The traditional Chinese explanatory models of illness, which include cosmic causes like karma, *feng shui*, and spirits or demons, still remain popular but have either been stigmatized as superstition or "scientized" and nationalized after the adoption of Western medicine (Palmer 2011). Buddhism aside, most Chinese ways of "doing religion" have not been meaningfully integrated into the secular healthcare field. Buddhist, Daoist, and other religious groups certainly do provide many important social services in Hong Kong, but there is only one recently-established Buddhist chaplaincy service, itself influenced by Christian CPE.

Third, there is not a strong "pastoral" tradition, *per se*, in Chinese religions. Most Chinese temples in Hong Kong are community-led organizations that lack the permanent, ordained clergy to provide such care. I am not suggesting that forms of "spiritual care" do not exist in Chinese culture—they certainly do—but rather, structural differences mean that these are lay-led, sporadic, and lack the wide range of responsibilities associated with chaplaincy. Many services are contracted from outside the religious community on a need basis, divided between specialized practitioners like ritualists, diviners, spirit mediums, and *feng shui* masters. The closest analogues to Christian pastoral counselors are probably Buddhist or Daoist spiritual advisers who provide moral and existential guidance. While no

---

[10]　This is the case even for those institutions founded by and for the Chinese populace: The Tung Wah Group of Hospitals maintains many temples and provides funerary services, but this has not translated into dedicated in-hospital religious care.

longer an elite luxury, these roles have generally not coalesced into clinically-based services. It is also unknown whether such caregivers attend to all "modalities" of Chinese religion, or are focused on philosophical discourse and personal cultivation (as they are in Western contexts).

Lastly, there is limited research on Chinese religious or spiritual needs and consequently few philosophical or theological paradigms to guide chaplains. Buddhism is a frequent topic, but it is most often discussed by Western practitioners, the otherwise informative *The Arts of Contemplative Care* (Giles and Miller 2012) being a good example. Contextual works focus primarily on cultural sensitivity toward ethnic Chinese, often framing Confucianism as a secular relational ethic (e.g., Lai 2003). These clearly do not address the full range of Chinese Buddhist or Confucian expressions. More religiously-oriented research comes from Hong Kong, Taiwanese, and Chinese/East Asian-diasporic medical and nursing studies (Mok et al. 2009; Tzeng and Yin 2006; Woo 1999), which confirm the need for Chinese spiritual care but provide only tentative guidance for providing it in the context of clinical work.

Based on the historical, anthropological, and sociological characteristics of Chinese religion, as well as the limitations of current research, we can conclude that a "Chinese spiritual care" is still a work in progress. In Hong Kong this creates an ambiguous situation: On the one hand, it is thanks to Christian influence and Western understandings of religious freedom that publically-offered forms of spiritual care like chaplaincy exist.[11] On the other hand, Christianity's privileged status raises ethical questions when it comes to spiritual care. Ultimately what is needed is a practical and theological paradigm for Christian chaplains that can accommodate Chinese belief. The absence of a clear Chinese religious alternative, however, makes this all the more urgent, which leads us back to the utility of the concept of hospitality.

## 8. Interreligious Ritual Hospitality as a New Paradigm

### 8.1. Hospitality in Therapy

An interfaith spiritual care informed by interreligious hospitality can be a new scheme or paradigm to overcome these obstacles. While not greatly explored with respect to interfaith chaplaincy[12], hospitality has received some attention in secular therapy. Bazzano (2015) proposes it as a form of radical ethics, informed by Levinas and Derrida, that can overcome a Cartesian notion of the self in therapeutic settings. Notably, the task of the therapist involves the same two epistemological and ethical questions that comparative theology and interreligious dialogue deal with, namely: (1) Who is "the Other"? and (2) how do we respond to them? Similar to Ricoeurean interreligious hospitality, the first involves surrendering the need to know and understand them, and the second calls for acting hospitable (Bazzano 2015). Rober and De Haene use the Derridaen impossibility of absolute hospitality to identify the ambivalence of compassion and violence in the therapeutic relationship (Rober and De Haene 2017). Such an understanding removes the unrealistic idealism and Western-centrism of "patient-centered" approaches.

Returning to comparative theology, an ethic of hospitality tries to understand another religion without demystifying it, but also keeps it mysterious without estranging oneself from it. This dialectic between familiarity and alterity is crucial for the East Asian and Chinese context, where religion has been reductively objectified through the "Western gaze." A comparative theologian (or a chaplain doing comparative theology) can be more acutely aware of the nuances, surprises, and even contradictions in Buddhism, Daoism, and Confucianism, as well as in the popular Chinese beliefs and practices that are frequently ignored or dismissed as non-religious or "superstitious." An excellent (if overused) example is the aversion to speaking about death among some Chinese families. To a chaplain, the

---

[11]  This is compared to Taiwan, where clinical chaplaincy does not seem to be as widespread, and Mainland China, where public religious services are restricted.

[12]  It is occasionally discussed as a caregiving virtue. See pt. 1, chap. 5 of *The Art of Spiritual Care* (Stratford 2016).

immediate meaning of refusing to say die or death (*si*) may be an irrational superstition, but a hermeneutically-open approach would reveal a range of higher meanings that would challenge the chaplain's previous understandings. Not only is it spirituality inauspicious, the taboo of death has deep cultural significance and psychological implications.

### 8.2. Comparative Theology and Ritual

While Moyaert recognizes the significance of Ricoeur's textual model of human action, she is critical of his "over-textualizing." By this she means not just his over-reliance on actual texts, but also seeing *everything* based on the *model* of the text. This model privileges the mind over the body, vision over the other senses, and freezes or reifies human action at a moment in time, rather than understanding human life as a dynamic and interactive process (Moyaert 2017). Of course, this is often unavoidable. Chaplains and chaplaincy departments need to justify their expense, and this means hard data is expected, prompting the growing need for evidence-based, outcome-oriented models of care. Practically-speaking, the patient and the spiritual action must always be "textualized" in some way as part of the hermeneutical-reflective process. Some sort of "spiritual assessment" must be made, and many US chaplains write directly onto their patients' charts, literally making them text. In CPE training, written accounts of spiritual care conversations ("verbatims") are used for group reflection. Capps' understanding of pastoral actions helps avoid this by reflecting on the whole event, rather than individual data points.

Moyaert has suggested revising Ricoeur's hospitality to include not just text or "textualized" actions, but religious *ritual* (Moyaert 2017). This approach reflects a turn in comparative theology from inter-texting to inter-*riting*, whereby dialogue is able to reach a more affective, experiential, and embodied level (Moyaert 2015).[13] Rituals are repeatable and symbolic practices that open up new worlds of meaning between them and the performers. The performers are detached (*distanciated*) from the actual temporal event, its specific structure, and their own original beliefs and reasons for performing it (Moyaert 2017). Similar to Yong's treatment of hospitable practices, a ritual focus offers an alternative to the intellectual truth-claims that limit TORs and other systematic theologies, instead seeking commonality in practice and performance.

Like a text, a shared ritual cannot have just *any* meaning, but must be validated vis-à-vis a number of conditions, including the different perspectives of the co-performers and an awareness of confessional boundaries, which again refers us back to theologies of religions. Mark Heim calls attention to the importance of personal intention and moral and spiritual intuitions in inter-ritual practice. In some rituals, intention is moot—just thoughtlessly performing a ritual is supposed to be effective. But for most ritual acts, the personal reasons behind the performance do matter. Intuitions are the immediate moral inclinations that a person may have regarding the idea of performing another religion's ritual. These are related to values shaped by a person's community and theology, such as loyalty and betrayal. For example, exclusivists may intuitively favor values like *authority* and *loyalty* to their traditions, while pluralists are concerned with fighting *oppression* or *unfairness* in interreligious dialogue (Heim 2015).

This means that personal theological convictions still matter very much. A ritual hospitality is not absolute hospitality, rather it is an ethic by which a theologian commits to being vulnerable, allowing comparative dialogue to challenge her deeply-held beliefs about "the Other" and about what is possible. There will still ultimately be limits to what she can do, but she at can least reflect upon and test these limits. Thus, while it starts as a practical ethic for facilitating dialogue, true hospitality—inter-textual or inter-ritualistic—is always hermeneutical and transformative: new worlds of meaning are opened up in the encounter.

---

13   This is from the introduction to a volume on inter-ritual participation that Moyaert co-edited, *Ritual Participation and Interreligious Dialogue* (2015). It also includes the essay by Mark Heim.

*8.3. Ritual and Interfaith Chaplaincy*

Chaplaincy can be a more concrete example of this hermeneutical tension. A chaplain would need to be discerning about whether or not she can participate in a patient's ritual, weighing her confessional commitments, theology, and being aware of her intentions and intuitions. Inter-riting may be more possible in the Hong Kong Chinese context, since ritual boundaries are delineated by pragmatic and functional concerns rather than confessional identity. Many patients would probably not have a problem receiving a Christian prayer, and some might even participate! The Christian chaplain, however, would certainly need to consider her own feelings on participating in a non-Christian practice. For instance, a more conservative practitioner may feel disloyal repeating the Buddhist Heart Sutra, or praying to Guanyin, a popular deity and bodhisattva in Hong Kong. Alternatively, the chaplain may personally reinterpret such things in a more philosophical or "secular" way that can be reconciled with Christian religious belief. This would unlikely conflict with the beliefs or intentions of the patient, as it is also a valid Buddhist understanding of the practice.[14] On the other hand, reinterpretations can be problematic. Chinese Christian converts face psychosocial pressure to perform or accept the practice of ancestral veneration. While not necessarily "spirit worship" *per se*, reinterpreting it as only secular remembrance or respect of the deceased may still cause conflict with more traditional family members.

Interreligious ritual hospitality is particularly appropriate to interfaith chaplaincy for several other reasons. First, it pertains to the more explicitly religious responsibilities of the chaplain, which can help them distinguish themselves from other clinical professionals. By being open to greater ritual involvement, a Christian chaplain has something besides friendly conversation to offer a non-Christian. Second, ritual *hospitality* overcomes—but does not ignore—the theological conflicts (i.e., competing views of truth, salvation, and human wellness) by resisting their tendency to cause *hostility*. Rather than seek an intellectual theological consensus, a ritual focus instead relies on the shared practices of hospitality. This also avoids the habit of resorting to abstract intellectual conversations of spirituality in interfaith chaplaincy encounters; While not insignificant, these may avoid a patient's real needs. Third, ritual helps "de-textualize" chaplaincy. As an embodied practice involving all five senses, ritual is more appropriate for the modern holistic approach to health. Rather than simply an emotional and mental exercise, ritual involves materiality and physicality. Ritual also cannot be reduced to simple visual and textual information, and thus requires the chaplain to commit to a narrative form of spiritual assessment, as opposed to merely checking-off boxes on a patient's chart. Finally, ritual is an especially useful way of looking at spirituality and religion in Hong Kong, where orthopraxy (right practice) is more important than orthodoxy (right belief).

Even if she decides she cannot directly participate in a ritual, a chaplain can at least facilitate it in other ways, chiefly by creating a safe, hospitable space for the patient to perform their own ritual, or even making a referral to an outside ritual expert. But this facilitation should not be passive, or else it would not be true hermeneutical hospitality. The chaplain should try to be present, witnessing the ritual and trying to understand it within the wider context of the patient's health. Relatedly, there is also more to ritual in spiritual care than the traditional acts of prayer, meditation, and sacraments. While I hesitate to call all human interaction ritual (that would render it meaningless), the more mundane aspects of chaplaincy are also ritualistic. I am thinking particularly of a chaplain's self-introduction, establishment of the conversational space, and leave-taking. These are things a chaplain will do in almost every visit, and there is a distinctive, repeatable pattern to them. Mirroring the chaplain's actions are the patient's own welcome (or rejection), collaboration in establishing the space, and leave-granting. These are not only hospitable rituals but rituals *of* hospitality, as they involve welcoming "the other" into one's presence.

---

14　For many Buddhists, theistic or devotional practices are a form of "skillful means" (*upaya*) which direct a person toward true non-theistic, non-dualistic enlightenment.

Therefore, a chaplain is rarely *just* facilitating ritual, but is also participating in a ritual relationship with the patient. While they may seem unimportant, pastoral visits, no matter how simple or mundane, can still be "meaningful actions", opening up a wider world of meaning that enables—and requires—reflection and re-reflection. It is certainly possible that there are visits that do not amount to much interaction, making significant hermeneutical work unnecessary. However, in many—if not most—encounters a chaplain performs some sort hospitality ritual. When genuine hospitality is involved, rather than mere "politeness" or "tolerance", interpretation is involved as well, and this will ultimately have an effect on the participants.

## 9. Conclusions

Not only does interreligious hospitality seek to balance the tensions inherent in comparative and interreligious work, it better maintains the "otherness" of other religions. By respecting alterity, the hospitable pastoral theologian and spiritual caregiver can avoid stereotyping or simplifying other religions, even as she seeks to understand them in the context of her own theology. An emphasis on praxis and performance overcomes the speculative problem of conflicting truth-claims among religions. Instead, through ritual hospitality, the caregiver can open up a hospitable space where their patient can safely express their spiritual beliefs, practices, and needs.

This does not ignore theological conflicts or even resolve them, but focusing on hospitality allows the act of care to proceed by resisting hostility. It is important to stress that this is not simply a deferral or "bracketing" of competing truth-claims—hospitality also implies vulnerability and hermeneutical openness on the part of the theologian or chaplain. By being interreligiously hospitable, they open their own deeply-held beliefs to the possibility of reinterpretation and transformation.

I have mostly discussed interreligious hospitality from the Christian perspective. This is grounded in a theological understanding of hospitality as an imperfect human expression of God's superabundant, unconditional hospitality, enabled through the sustaining and creative presence of the Holy Spirit. That said, interreligious hospitality can be a hermeneutic for scholars and practitioners of other faiths, but it will need to account for the differences in philosophy, ethics, and practices of hospitality in those contexts. Going forward, a truly interfaith understanding of pastoral hospitality will require dialogue with the hospitalities of other religions.

**Funding:** This research received no external funding.

**Conflicts of Interest:** This author declares no conflict of interest.

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
