# Peer review of "Interfaith Chaplaincy as Interpretive Hospitality"

_religions, doi:10.3390/rel10030226_

Round 1
Reviewer 1 Report
Please see all my comments on the manuscript that is attached.
Excellent work!
Thank you

Author Response
To the Reviewer. Many thanks for your helpful comments. I have addressed them point by point below. Several sections were highlighted without comment (as far as I could tell). I indicated these as well, just in case you had made some comments that were deleted by accident. If revisions were needed in these cases, please let me know.
Point 1: I agree. Offering spiritual care in healthcare settings can create some tensions for the chaplain. But the tensions transform into creative tensions as the context assists us in the process of translating the text (person).
Response 1: Excellent point. I have added a positive point regarding tension at the beginning of paragraph 2.
Point 2: This text in the 1st paragraph of section 2 was highlighted. "Even pluralism must logically exclude more conservative exclusivist positions"
Response 2: No comment was included on the document, but I wanted to make sure there wasn't a suggested revision.
Point 3: The author makes a significant statement here that all major faith groups shows sacred commitment to the stranger. This assists the chaplain along with other caregivers in a pluralist environment that radical hospitality is not just a core value of Christianity.
Response 3: Thank you for the comment. As (I hope) I indicated in the conclusion, non-Christian understandings of hospitality are an area that needs to be explored.
Point/Response 4: Replaced "spirituality" with "spiritually"- thanks for catching it.
Point 5: This passage from section 4 was highlighted, but with no comment: "A “particularist” believes that these systems are so unique that they are unknowable (untranslatable) by those from other religions, which makes meaningful interreligious dialogue impossible. While she affirms that cultural-linguistic models are useful, Moyaert rejects this narrow view, arguing instead for a form of comparative theology that remains open to the strangeness of other religions. Theologically, Moyaert grounds her argument in a reinterpretation of the Story of Babel (Gen 11), which she views not as humankind’s “second fall”, but as an act of God affirming cultural and linguistic diversity. Babel was the start of linguistic polysemy—in which words or phrases have many different meanings (Moyaert 2011)."
Response 5: No comment was included, but I wanted to make sure there wasn't a suggested revision.
Point 6: I recommend that the author include readings from Thomas E. Reynolds, especial his book, Vulnerable Communion: A Theology of Disability and Hospitality. Reynolds, Amos Yong, and others in the field of Disability Theology brings a wealth of knowledge regarding hospitality. As a Chaplain serving a community of 200 residents, their families and staff who have disabilities, I have learned the gift of hospitality by them. Their vulnerable state via social stigmatization, deems them vulnerable, although they are less vulnerable within our community. They have taught me that to be hospitable one needs to allow being vulnerable. This has been a key learning for me which has affected my embedded theology for positive change.
Response 6: Thank you for bringing this text to my attention, as I had not yet come across it. I feel I should read it thoroughly before feel I can confidently cite it, but it will show up in future work. However, I have made sure to stress in a couple of places that vulnerability is a critical element of Christian practice.
Point 7: I'm glad Donald Capps is included in the article. Capps includes the larger picture of the pastoral interventions regarding process and several interventions. Also, it might be notable to include the founder of CPE, Antone Boisen, as he brought forth going forward the hermeneutic significance of the action-reflection-action model from a theological and psychological basis.
Response: This is correct, so I mentioned Boisen as well.
Point 8: This is a significant example from the author's CPE experience to illustrate the range of possible patient responses.
Response 8: Thank you.
Point/Response 9: I corrected this typo--thanks for catching it.
Point 10: It is precisely in the more mundane times that meaning-making takes place within a continuity of care. It is in the process of careful and mutual reflection and re-reflection that the caregiver and care-receiver-seeker develop a meaningful relationship. This relationship then provides the backdrop for meaningful care.
Response 10: I have reworked this segment to reaffirm the importance of the "mundane".
Point 11: See Dykstra, Robert - In his book, Images of Pastoral Care more images can be added besides shepherd, wounded healer, and wise fool. Also, given the author's acceptance that these images perhaps are outdated and perhaps narrowly drawn from just one religious community, I would recommend the author including other helpful and pastoral images evolving from other religious entities.
Response 11: This is a fair point. I have rewritten this paragraph, adding a bit about Dykstra and omitting "outdated" (it wasn't necessary or fair). This led me to expand a bit on the paradigms or "images" of spiritual care.
Point 12: I like use of the term Christian interfaith chaplains. Being on the field with other Christian chaplains, there seems to be a tension with being a Christian and serving interfaithly. This term reminds us that being a Christian (follower of Jesus Christ) we can provide spiritual care with non-Christians as well as with Christians.
Point 12: Actually, my intent had been to be clear as to my audience (Christian interfaith chaplains), but you raise another good point about affirming identity while also serving the "wider Church".
Point 13: This section was also highlighted, but without comment"
"Most Christian chaplains are not well-enough equipped or informed to code-switch between religions, and doing so risks breaching sacred boundaries and rules. The exception may be those beliefs, texts, or practices shared in common, such as a Christian and a Jew reading a Psalm together, or a mystical discussion between a Christian and a Muslim. But besides this, there are few meaningful “generic” spiritual services the Christian chaplain could provide the Muslim patient. While they might think “everyone is spiritual” this would ignore a Muslim’s very specific and concrete religious needs (Abu-Ras and Laird 2011). Worse, this cuts very close to cultural or religious appropriation, even if the intent is to help. This is why Moyaert thinks Ricoeur’s third step of “appropriation” is better understood as expropriation. The former implies “interpretive imperialism”, while the latter better expresses a linguistic hospitality where the other religion is kept unfamiliar, but the theologian’s religious beliefs and practices are “redescribed and transformed” (Moyaert 2017, p. 180)."
Response 13: Just checking to make sure there aren't any problems or suggestions with this highlighted section.
Point 14: See, Kraus, Paul. Increasing Critical Self Awareness, a Revised Model for Pastoral Care, (www.tren.com Theological Research Exchange Network) 2008. In this study there is a careful and sensitive understanding between Individualistic and Collective Cultures as it pertains to spiritual/pastoral care education. Person-centered care and independence are in tune with individualistic values, but not so much from collective cultures. There seems to be a tension with this especially within western medicine and western healthcare systems. Patient-centered/person-centered care is central to care in western medicine. However, this is much better than paternalistic care which was the case early on in medicine. I believe there should be a communal or collaborative role in healthcare and it should be included in the sphere of the person-centered care.
Response 14: I haven't yet found this work, but I will be sure to try and include it in future research. A more comprehensive explanation of Chinese spiritual needs is the focus of another paper.
Point 15: There is definitely an on-going tension with chaplains to preserve their own faith as they support another person's faith during their interfaith service of another.
Response 15: Thank you...I strengthened this aspect a little bit.
Point 16: The author makes a very good point here especially using the example of confronting a "patient" with the word death or dieing. For the western Christian chaplain, to not confront the patient with this kind of reality would be a death denial concern. Not so with Eastern-non-Christians.
Response 16: Thank you.
Response 17: This section 6 passage was highlighted with comment: "Chaplains and chaplaincy departments need to justify their expense, and this means hard data is expected, prompting the growing need for evidence-based, outcome-oriented models of care. Practically-speaking, the patient and the spiritual action must always be “textualized” in some way as part of the hermeneutical-reflective process. Some sort of “spiritual assessment” must be made, and many US chaplains write directly onto their patients’ charts, literally making them text. In CPE training, written accounts of spiritual care conversations (“verbatims”) are used for group reflection. Capps’ understanding of pastoral actions helps avoid this by reflecting on the whole event, rather than individual data points."
Response 17: Just checking...any recommended changes?
Point 18: Often times the chaplain refers the rituals/practices to other clergy outside the hospital. The chaplain becomes the conduit connecting other clergy with the patient. But in order for this connect-ability to happen the chaplain needs to nurture the professional relationship the clergy that surrounds the hospital. The chaplain's success is evident per clergy responding to a call even in the middle of the night.
Response 18: Indeed. This is a critical issue especially for the Hong Kong context, where I suspect such arrangements are not necessarily understood by the local religious leaders. I will expand on this in the future.
Point 19: Here, the author invites the chaplain to participate in rituals that are not usually apart of the chaplain's ritual. I think this is just as important as referring clergy to do the ritual. Being apart of the ritual gives evidence of hospitality.
Response 19: Yes, I think a chaplain should challenge oneself to push against ritual boundaries, demonstrating hospitality. I have re-emphasized that throughout the paper.
Point 20: In this age of data entry its important that chaplains chart in narrative form opposed to "checking-off boxes". Additionally, I suggest that the chaplain provide vignettes (outside of the CPE verbatim and case study requirements). Vignettes (with proper use of confidential verbiage) can be used to share with hospital administrators who are interested in taking a peek at what the chaplain offers patients and their families.
Response 20: Absolutely! Verbatims are good for self-reflection, but such teaching moments can be helpful for the rest of the interdisciplinary team.
Point 21: I often share with my students that our mere presence is a sort of prayer ritual.
Response 21: Here I put "mundane" in quotes in order to emphasize that I think there is nothing really mundane about patient-chaplain interaction. I added more explanation about the importance of "presence", as I think I needed to reaffirm that I still think it is a crucial part of chaplaincy.
Point 22: I firmly agree!
Response 22: I hope to tackle the hospitalities of other religions soon.
Reviewer 2 Report
It is a very interesting and relevant topic that the author approaches with originality. The general thinking that conversation is less a problem in interreligious care than ritual is subtly challenged by a plea for interreligious ritual hospitality as a facilitating practice.
At the same time, the clarity of writing almost hides some lack of clarity in regard to the major strands of thought. If I read correctly the author seeks to bring together two major strands of thought. One is the conceptualization of hospitality as an interreligious practice that sidesteps some of the confrontations of theologies of religious pluralism. The other is the value of ritual as a a material, physical practice. The bringing together needs, in my view, some more critical reflection to understand what is happening in the proposed hermeneutical hospitality. The author signals a move from hermeneutics of text (Ricouer) to hermeneutics of human interaction (Moyaert) to a hermeneutics of ritual, but hardly discusses, especially in relation to the last move, the real difficulties involved. And is ritual hospitality really hermeneutical or just facilitating? So how does the hermeneutics of it fit in? And the author claims that: "An emphasis on praxis and performance really overcomes the speculative problem of conflicting truth-claims among religions." (402) What kind of claim is that in itself? And are the truth-claims overcome, or side-stepped as I suggested above? I would like to invite the author to explicate some of these things.
There are in addition some other less severe issues:
Twice the author introduces responses of others to MacIntyre 'meta-ethics' without explaining what that is and without referencing it. (66, 103
The distinctions exclusivist, inclusivist and pluralist are said to have been much debated and deconstructed but they are nevertheless employed. Why? 51-52, 139, etc.
There are passages (116, 141) that might seem to relativize the radical tension that Derrida attributes to hospitality (85-90, 105, 110) but that is not acknowledged.
The author uses his/her own Asian context to confront some of the literature. Almost all of the literature is Western. Is there not other literature to cite from the Asian context?
In short, I think the article is good reading, addresses vital issues, but should be more clear on some of the difficulties involved.
Author Response
These were incredibly helpful comments, and were the basis of a few significant revisions. Many, many thanks. I hope the paper reads more clearly now. I've address your points below:
Point 1: The bringing together needs, in my view, some more critical reflection to understand what is happening in the proposed hermeneutical hospitality. The author signals a move from hermeneutics of text (Ricouer) to hermeneutics of human interaction (Moyaert) to a hermeneutics of ritual, but hardly discusses, especially in relation to the last move, the real difficulties involved.
Response 1: Excellent critique. I have tried to better address the difficulties, particularly the fact that in human interaction and ritual boundaries will need to be set up to avoid offense or betrayal of the chaplain's own beliefs (if that was your meaning). In one place I make reference to the difficulty of ritual participation with respect to Chinese Christians. These difficulties are further address with respect to your specific questions:
Point 2: And is ritual hospitality really hermeneutical or just facilitating? So how does the hermeneutics of it fit in?
Response 2: I added several sentences regarding this good question. See in particular the new paragraph at the end of section 8, subsection 2:
This means that personal theological convictions still matter very much. A ritual hospitality is not absolute hospitality, rather it is an ethic by which a theologian commits to being vulnerable, allowing comparative dialogue to challenge her deeply-held beliefs about “the Other” and about what is possible. There will still ultimately be limits to what she can do, but she at least reflects upon and tests these limits. Thus, while it starts as a practical ethic for facilitating dialogue, true hospitality— inter-textual or inter-ritualistic—is always hermeneutical and transformative: new worlds of meaning are opened up in the encounter.
As well as at the end of section 8:
Therefore, a chaplain is rarely just facilitating ritual, but is also participating in a ritual relationship with the patient. While they may seem unimportant, pastoral visits, no matter how simple or mundane, can still be “meaningful actions”, opening up a wider world of meaning that enables—and requires—reflection and re-reflection. It is certainly possible that there are visits that do not amount to much interaction, making significant hermeneutical work unnecessary. However, in many—if not most—encounters a chaplain performs some sort hospitality ritual. When genuine hospitality is involved, rather than mere “politeness” or “tolerance”, interpretation is involved as well, and this will ultimately have an effect on the participants.
Here I claim that, at minimum, hospitality facilitates. However if it is genuine hospitality it will include vulnerability and hermeneutical openness, ergo it will affect and possibly transform the the interpreter's views.
Point 3: And the author claims that: "An emphasis on praxis and performance really overcomes the speculative problem of conflicting truth-claims among religions." (402) What kind of claim is that in itself? And are the truth-claims overcome, or side-stepped as I suggested above? I would like to invite the author to explicate some of these things.
Response 3: I added explanation in a couple of places, most importantly the conclusion, stressing that overcoming is not merely side-stepping. The conflict between truth-claims remain and will affect how individuals interact, often encouraging hostility. Rather, per my Response #2, hospitality refers to a commitment to be vulnerable and open to the stranger despite this, though of course there will be limits.
Point 4: Twice the author introduces responses of others to MacIntyre 'meta-ethics' without explaining what that is and without referencing it. (66, 103)
Response 4: As I only wished to emphasize Bretherton's critique of meta-ethical paradigms, I ultimately decided it was not necessary to include MacIntyre in the body of the paper. Instead, I refer to him in the footnote.
Point 5: The distinctions exclusivist, inclusivist and pluralist are said to have been much debated and deconstructed but they are nevertheless employed. Why? 51-52, 139, etc.
Response 5: I added explanation that these are problematic, but ultimately they represent a spectrum of perspectives upon which any theologian or practitioner will fall (i.e. everyone will be inclined toward one of these, regardless). I cited Yong here, at the opening paragraph of section 2, as he holds the same opinion.
Point 6: There are passages (116, 141) that might seem to relativize the radical tension that Derrida attributes to hospitality (85-90, 105, 110) but that is not acknowledged.
Response 6: Thank you, I have tried to better emphasize that it that any kind of hospitable practice will fall short of Derrida's absolute hospitality, ergo the paradox holds.
Point 7: The author uses his/her own Asian context to confront some of the literature. Almost all of the literature is Western. Is there not other literature to cite from the Asian context?
Response 7: In my view, there is very little literature directly related to hospitality and interfaith care from the Chinese perspective, but you raise a good point, insofar I had not really established that. I have added an entire section on Chinese religion and spiritual care needs, stressing that the lack of research is one of the reasons why I am writing this.
Furthermore, I reemphasized that I am writing from a Christian interfaith perspective. Buddhist, Daoist, Confucian, etc. models of care and/or hospitality are needed (in the very least an understand of Chinese religion/spirituality), but ultimately this is hospitality framed from a Christian POV, ergo the Christian emphasis.
One clear blind spot is a Chinese Christian theological perspective, especially the comparative theological kind. That could be part of a longer discussion.
Again, many thanks